# Biomolecular condensate phase diagrams with a combinatorial microdroplet platform

William E. Arter [1,9], Runzhang Qi [1,9], Nadia A. Erkamp[1,9], Georg Krainer [1,9], Kieran Didi [1], Timothy J. Welsh [1], Julia Acker [2], Jonathan Nixon-Abell [3], Seema Qamar[3], Jordina Guillén-Boixet[4], Titus M. Franzmann [4], David Kuster [5], Anthony A. Hyman [5], Alexander Borodavka[2], Peter St George-Hyslop [3,6,7], Simon Alberti [4] & Tuomas P. J. Knowles [1,8] ✉

The assembly of biomolecules into condensates is a fundamental process underlying the organisation of the intracellular space and the regulation of many cellular functions. Mapping and characterising phase behaviour of biomolecules is essential to understand the mechanisms of condensate assembly, and to develop therapeutic strategies targeting biomolecular condensate systems. A central concept for characterising phase-separating systems is the phase diagram. Phase diagrams are typically built from numerous individual measurements sampling different parts of the parameter space. However, even when performed in microwell plate format, this process is slow, low throughput and requires significant sample consumption. To address this challenge, we present here a combinatorial droplet microfluidic platform, termed PhaseScan, for rapid and high-resolution acquisition of multidimensional biomolecular phase diagrams. Using this platform, we characterise the phase behaviour of a wide range of systems under a variety of conditions and demonstrate that this approach allows the quantitative characterisation of the effect of small molecules on biomolecular phase transitions.

Many cellular protein and nucleic acid molecules have the ability to demix into a dense liquid condensate phase and a dilute phase depleted of biomolecules [1]. This liquid–liquid phase separation (LLPS) process, has emerged as a fundamental mechanism to describe the formation of biological condensates in living cells[2]. Condensate systems such as stress granules, paraspeckles, and nuclear bodies are formed by LLPS and organise and compartmentalise cellular space[3]. They act as microreactors for biochemical reactions[4], and are crucial to a wide range of physiological processes such as gene expression,

signalling, and metabolic regulation[5–7]. Condensate-forming proteins are also heavily implicated in protein misfolding diseases including motor neurone disease[8–10], cancer pathogenesis[11] and infectious diseases[12], making them attractive targets for therapeutic intervention[13,14].

Phase separation, now established for hundreds of cellular proteins[15], can be triggered by variations in environmental conditions such as changes in ionic strength, pH, temperature, molecular crowding, and the presence of small molecules[16–21]. It is important that

[1]Yusuf Hamied Department of Chemistry, Centre for Misfolding Diseases, University of Cambridge, Lensfield Road, Cambridge CB2 1EW, UK. [2]Department of Biochemistry, University of Cambridge, Cambridge CB2 1QW, UK. [3]Cambridge Institute for Medical Research, Department of Clinical Neurosciences, University of Cambridge, Cambridge CB2 0XY, UK. [4]Biotechnology Center (BIOTEC), Center for Molecular and Cellular Bioengineering (CMCB), Technische Universität Dresden, Tatzberg 47/49, 01307 Dresden, Germany. [5]Max Planck Institute for Molecular Cell Biology and Genetics, Pfotenhauerstrasse 108, 01307 Dresden, Germany. [6]Department of Medicine (Division of Neurology), University of Toronto and University Health Network, Toronto, ON M5S 3H2, Canada. [7]Department of Neurology, Columbia University, 630 West 168th St, New York, NY 10032, USA. [8]Cavendish Laboratory, Department of Physics, University of Cambridge, J J Thomson Ave, Cambridge CB3 0HE, UK. [9]These authors contributed equally: William E. Arter, Runzhang Qi, Nadia A. Erkamp, Georg Krainer. ✉e-mail: tpjk2@cam.ac.uk

cellular concentrations of condensate components are considered during in-vitro analysis, given the high sensitivity of cellular phase separation to parameters such as protein concentration[22]. Because of this sensitivity to physicochemical parameters, there is great potential to modulate protein phase separation in a targeted manner and many phase-separating proteins and their complexes are receiving intense interest as a new class of drug targets with which to treat human diseases[23,24]. However, it remains challenging to quantify the physical parameters that modulate phase separation behaviour in LLPS systems with high fidelity, accuracy and throughput[16].

A central concept that characterises phase-separating systems is the phase diagram[24–26]. Phase diagrams summarise the phase behaviour of condensates by monitoring the position of the transition between the phase-separated and the mixed regimes through the determination of the position of the phase boundary in chemical space[27]. Changes in the phase boundary for an LLPS system following variation of solution conditions affords insights to the thermodynamic processes driving protein condensation and the factors that modulate them[17]. However, given the large variety of proteins undergoing LLPS and the environmental conditions which regulate their behaviour, there is a need for experimental methods that enable rapid and high-resolution characterisation of LLPS phase diagrams. Typically, these are generated by time intensive methods involving the stepwise combination of reagents to create the requisite variation in solution conditions before the observation of individual conditions by microscopy. Notably, although phase diagrams are often used to probe the behaviour of LLPS systems, such experiments can only explore a limited set of conditions and/or provide a coarse-grained picture of the protein phase-space when conducted manually[28,29].

To address this challenge, we describe here a droplet microfluidic approach, PhaseScan, that enables rapid, automated generation of different LLPS solution conditions for high-resolution assessment of protein LLPS behaviour. Lab-on-a-chip microfluidic systems, in particular microdroplet-based formats, offer an effective means to improve assay throughput, parallelisation, and miniaturisation in biochemical experiments. To date, these platforms have been leveraged to probe protein phase behaviour[30–33] by concentration or evaporation of dilute, homogeneous protein solutions contained in microwells or microdroplets, so that the system becomes sufficiently concentrated to pass into the phase-separated regime. Initiating phase separation by concentrating the solution means that salt and buffer are also concentrated, making it challenging to probe phase diagrams under fully physiological conditions or to map out phase diagrams without chemical or physical dimensions being interdependent.

We aim to overcome these challenges through combinatorial droplet microfluidics to rapidly generate a large number of independent micro-compartments, each of which can be considered a discrete environment in which to study protein LLPS. This approach allows a far greater number of unique conditions to be probed than alternative microfluidic approaches where assay components are passively mixed and compartmentalised by the fluidic device itself. Such formats limit the number of distinct solution conditions that can be assayed by the extent and complexity of the fluidic network that is possible and practical to operate[33–35]. PhaseScan, by contrast, alters the input solution conditions in a combinatorial manner during droplet encapsulation, allowing for rapid generation of an arbitrary number of phase separation microenvironments to map LLPS behaviour over a broad range of chemical space. We demonstrate the operation of the PhaseScan platform by acquiring phase diagrams for a variety of phase-separating proteins and their modulation by molecular crowding, salt concentration, and nucleic acid concentration. We then investigate and compare the effect of small molecules on the protein phase separation. Finally, we show the acquisition of a three-dimensional phase diagram, demonstrating the potential of our platform for

parallelised, multi-dimensional analysis of chemical space in the context of protein phase separation.

## Results and discussion
### Working principle of the PhaseScan platform

To begin, we implemented a workflow for microdroplet generation and imaging (Fig. 1a). As a model system, we utilised an EGFP-tagged G156E mutant of the protein fused in sarcoma (FUS[G156E]), a protein implicated in the pathology of amyotrophic lateral sclerosis (ALS) that has previously been shown to phase separate[9,10]. Phase separation was triggered by molecular crowding with polyethylene glycol (PEG), as reported previously[9].

Aqueous solutions containing protein, buffer, and the modulator PEG were mixed in different ratios on chip prior to encapsulation in water-in-oil microcompartments of ~700 pL in volume dispersed in an immiscible fluorinated-oil continuous phase (Fig. 1b). Due to laminar flow, the aqueous solutions do not mix significantly before droplet formation[36], and droplet coalescence is prevented by supplementing the fluorinated oil phase with a biologically compatible polyglycerol-based triblock surfactant (see Methods, Droplet Generation)[37]. The relative flow rates of the aqueous solutions are altered while the total flow rate is kept constant using a programme controlled by an automated syringe pump system. This enables the formation of droplets containing protein and PEG concentrations over a range of chemical space. To determine the amount of protein and PEG present in each droplet in downstream analysis, the concentration of each of these components was barcoded by a particular fluorophore (Fig. 1b, c). For FUS[G156E], this information was provided by its EGFP tag, whereas the concentration of PEG was barcoded by Alexa647 dye pre-mixed into the PEG starting solution (see Methods, Droplet Generation).

Following generation, droplets were collected in a PDMS gravity-trap device, that utilises the lower density of droplets in comparison to the surrounding oil to immobilise droplets in microwells contained in the roof of a flow chamber, as described previously[30,38]. Approximately 2600 droplets were collected per experiment, with droplet generation occurring over a timescale that allowed at least two repetitions of the flow programme to ensure that the full range of conditions were scanned, before undergoing epifluorescence microscopy imaging according to the fluorescence wavelengths of each of the barcode fluorophores (Fig. 1d, e). According to the barcode fluorescence intensity, the corresponding concentrations of protein or PEG were determined on a per-droplet basis by comparison to a calibration measurement of known fluorophore concentration (see Supplementary Note 6 and Supplementary Fig. 9).

Droplets were then classified as phase-separated or homogeneous according to the presence or absence of condensates visualised via the protein EGFP tag (see Supplementary Note 7 and Supplementary Fig. 10). Figure 1g, f demonstrates the assignment for a representative image of phase-separated and homogeneous droplets containing EGFP-FUS[G156E], with a red or blue outline indicating the classification of droplets as phase separated or homogeneous, respectively. Typically, the sum of false-positive and false-negative classification error by our algorithm is <4% of the total number of droplets classified, although this value varies slightly depending on the signal-to-noise ratio of condensate fluorescence for the system in question.

We observed the merging of condensates over time (Fig. 1h), which together with off-chip fluorescence recovery after photo-bleaching (FRAP) experiments (see Supplementary Note 1 and Supplementary Fig. 1) confirmed the liquid nature of condensates formed in the PhaseScan experiment. By combining the determined concentration and the presence or absence of phase separation as a scatter plot on a per-droplet basis, a phase diagram was produced (Fig. 1i). The position of the phase boundary and probability of phase separation as a function of EGFP-FUS[G156E] and PEG concentration was

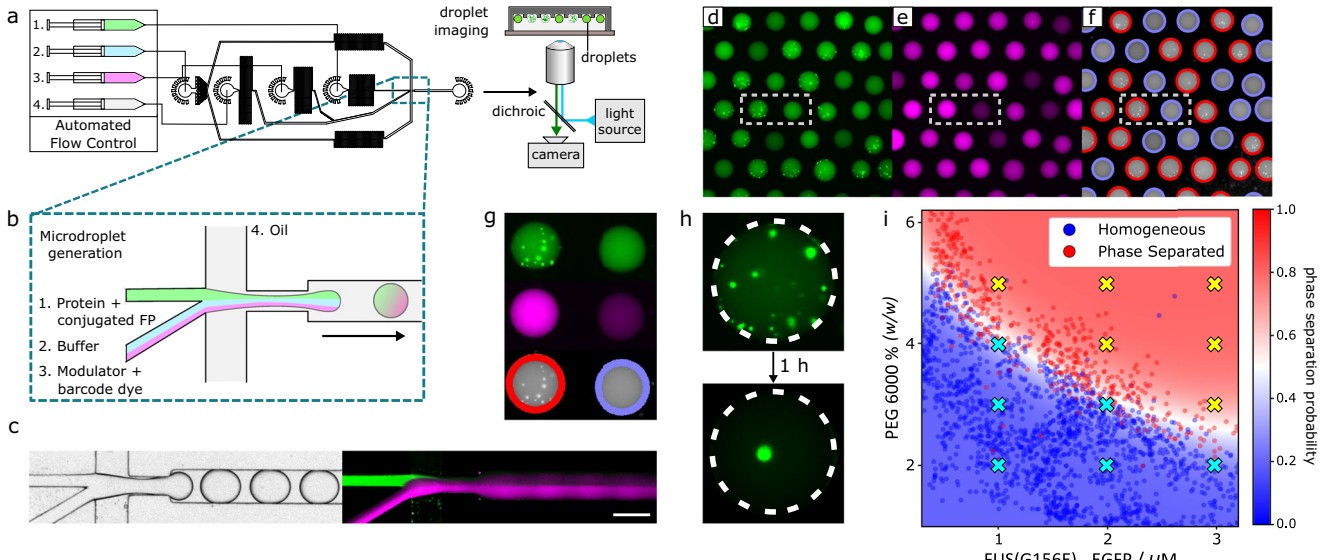

**Fig. 1 | PhaseScan workflow. a** Droplets are generated using a flow-focussing microfluidic device controlled by automated syringe pumps and then imaged in wells by fluorescence microscopy. **b** At the droplet generating junction, aqueous solutions are combined under laminar flow before droplet formation. **c** Brightfield microscopy image of droplet generation (left) and combined fluorescence images of droplet generation (right) showing fluorescence of EGFP (green) and Alexa647 (magenta) barcodes for FUS[G156E] and PEG, respectively. **d**, **e** Epifluorescence microscopy images of trapped microdroplets, with EGFP and Alexa647 fluorescence corresponding to FUS[G156E] and PEG concentration, respectively. **f** Classification of droplets as phase separated (red outline) or homogeneous (blue outline) according to distribution of EGFP fluorescence. **g** Phase separated (left) and homogeneous (right) microdroplets imaged according to EGFP (top) and Alexa647 fluorescence (middle) and subsequent phase separation classification (bottom). Images correspond to the highlighted regions in (**d**–**f**). **h** Liquid condensates merge over time in microdroplets. **i** Phase diagram of EGFP-FUS[G156E] vs. PEG 6000 concentration, 50 mM Tris pH 7.4, 150 mM KCl. Red and blue data points in the scatter plot correspond to individual microdroplets classified as phase separated or homogeneous, respectively. The heat map corresponds to the probability of phase separation as determined by an SVM classifier trained on the droplet scatter plot. $N = 2754$ droplets. Yellow and cyan crosses correspond to phase separated and homogeneous behaviour as determined by manual pipetting experiment. Source data are provided as a Source Data file. Parts of this figure are reproduced with permission from Geiger et al.[12].

then determined by a support-vector machine (SVM) algorithm trained on the scatter-plot populations of phase-separated or homogeneous droplets (Fig. 1i, see Methods, Droplet Detection and Phase Diagram Generation). Notably, only a technique with sufficient throughput allows the use of statistical methods to comprehensively and continuously describe phase behaviour in the manner demonstrated here.

We observe that droplets can be classified robustly into phase-separated or homogeneous regions according to concentrations of FUS[G156E] and PEG that show excellent agreement with previous studies[9]. To assess whether the phase behaviour of FUS[G156E] was notably altered by droplet encapsulation, the presence or absence of phase separation under the same conditions as those probed by PhaseScan were confirmed as the same in bulk volumes by manual pipetting experiments (Fig. 1i, Supplementary Note 2 and Supplementary Fig. 2). Importantly, although the phase-separation systems described herein are characterised by use of fluorophore-tagged proteins, this is not a pre-requisite for the operation of the PhaseScan platform as the presence of condensates within droplets could instead be observed by brightfield microscopy as shown elsewhere[30,33] and in Supplementary Fig. 1.

To test whether the barcoding dyes affect the phase boundary, the phase behaviour of FUS[G156E] was assessed as a function of Alexa546 and Alexa647 dye concentration (see Supplementary Note 3 and Supplementary Fig. 3). No significant effect was observed, with phase separation propensity remaining unchanged for the barcode concentrations used here (<6 μM, see Methods, Droplet Generation).

Phase-separated systems behave dynamically, with condensate droplets evolving after nucleation due to growth, coalescence and Ostwald ripening[39,40]. To test whether these processes effect the accuracy of the PhaseScan measurement, phase diagrams for homotypic FUS phase separation were generated in triplicate, with droplet generation proceeding continuously but with 30 min between data

acquisition for each replicate measurement. Only negligible differences were observed between the replicates (see Supplementary Note 4 and Supplementary Fig. 4), demonstrating that the PhaseScan protocol described here (>5 min between drop generation and imaging) allows sufficient time between droplet generation and measurement for the assay to accurately and reproducibly assess the equilibrium phase boundary position. This is in agreement with previous findings, where the characteristic timescale for condensate formation within droplets of >100 nL was found to be <1 min[41].

We also investigated the potential for droplet size to affect the reported position of the phase boundary, since droplet volume has the propensity to modulate the dynamics of nucleation-driven condensate growth[41]. Phase diagrams for homotypic FUS[G156E] condensates were recorded using droplets of two different sizes with mean volumes of 0.33 and 1.1 nL, which bracketed the typical droplet volume of 0.65 nL used in PhaseScan experiments (see Supplementary Note 5 and Supplementary Fig. 5). No difference was observed between these phase diagrams, which together with the lack of time-dependence in the PhaseScan output described above, indicates that the PhaseScan experiment reports the equilibrium position of the phase boundary and is insensitive to droplet volume.

Notably, the PhaseScan technique provides throughput several orders of magnitude higher than manual experiments in terms of the number of unique conditions investigated, with >2500 droplets assayed per experiment. Reagent consumption is minimal, with <2 μL of stock protein solution used per assay, although we found that a minimum working volume of 10 μL was required for effective operation (see Methods, Droplet Generation). Moreover, data generation is rapid, with droplet generation and collection requiring ~5 min. This is in contrast to microfluidic approaches for the investigation of LLPS based on droplet shrinking, which can require experimental timescales of several hours.

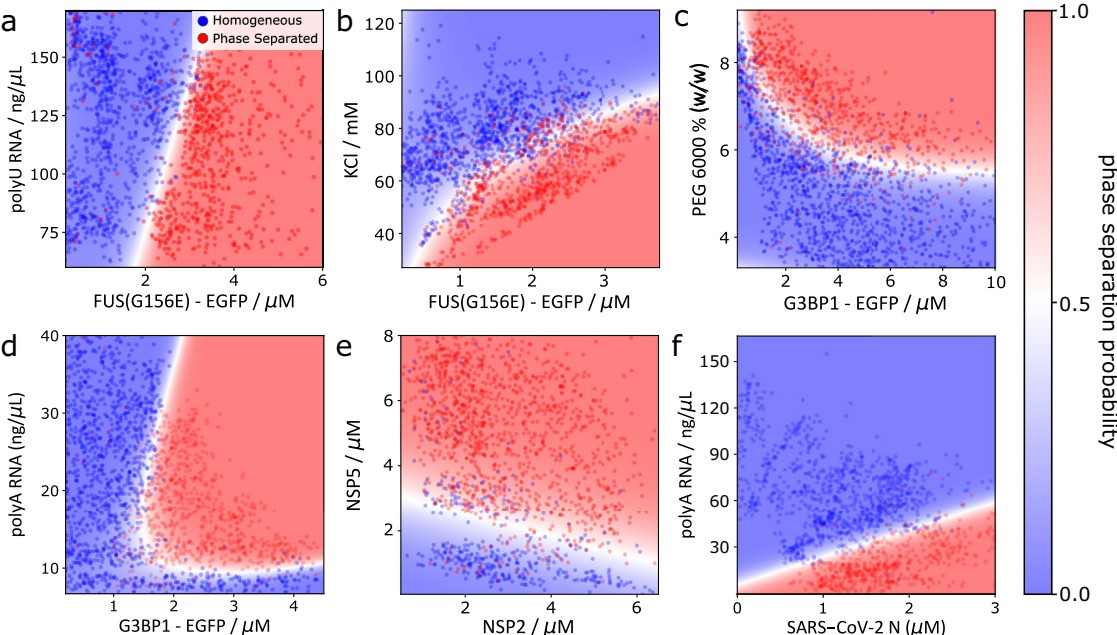

**Fig. 2 | Application of PhaseScan to a variety of condensate systems. a** Phase diagram of EGFP-tagged FUS[G156E] vs. polyU RNA concentration. $N = 2096$ droplets. **b** Phase diagram of FUS[G156E] condensation as a function of protein and salt concentration. $N = 2625$ droplets. **c** Phase diagram of EGFP-tagged G3BP1 vs. PEG 6000 concentration. $N = 2549$ droplets. **d** Phase diagram for polyA RNA-mediated phase separation G3BP1. $N = 3077$ droplets. **e** Phase diagram for coacervation-condensation of rotavirus proteins NSP2 and NSP5. $N = 1672$ droplets. **f** Phase diagram for polyA RNA-mediated phase separation of SARS-CoV-2 N protein. $N = 1599$ droplets. Red and blue data points in scatter plots correspond to individual microdroplets classified as phase separated or homogeneous, respectively. The heat map corresponds to the probability of phase separation as determined by an SVM classifier trained on the droplet scatter plot. Source data are provided as a Source Data file.

## PhaseScan is generalisable to a range of phase-separating systems

Having validated the operation of the PhaseScan system, we then sought to utilise our platform for the investigation of a range of phase-separating systems and test the generality of the method for homotypic and heterotypic condensate systems.

Since coacervation of nucleic acids is a defining characteristic of phase-separated ribonucleoprotein granules, we began by producing a phase diagram for condensation of EGFP-tagged FUS[G156E] in the presence of polyU RNA (Fig. 2a). As observed previously[42], no phase separation was detected at low ratios of FUS[G156E] relative to RNA, with phase separation then occurring at higher FUS[G156E] concentrations. Notably, the phase boundary appears highly linear, in agreement with studies that suggest that condensate formation and protein−RNA coacervation occurs via charge neutralisation[42], and that condensation therefore occurs at a discrete ratio of RNA to protein.

Next, we generated a phase diagram for the phase separation of FUS[G156E] as a function of KCl concentration (Fig. 2b). In agreement with previous findings, FUS[G156E] was observed to possess an increased propensity for phase separation at high protein and low salt concentrations. Analysis of the effect of ionic strength on the propensity of condensate systems to phase separate can afford mechanistic insight into the molecular-level interactions that drive LLPS, by elucidating the extent to which phase separation, for example, is influenced by electrostatic interactions[43,44]. We therefore envisage that PhaseScan could enable rapid, high-resolution mechanistic analysis of phase separation processes.

To demonstrate the applicability of the PhaseScan system to other phase-separating proteins, we next characterised the crowding-driven phase separation of EGFP-tagged G3BP1, a scaffolding protein required for the formation of stress granules (Fig. 2c)[45]. As expected, phase separation of G3BP1 was present at high concentrations of PEG molecular crowder, with this effect enhanced at higher protein concentrations.

We then applied the PhaseScan assay to study reentrant phase behaviour, a central concept in the LLPS field. The formation of ribonucleoprotein condensates is largely driven by favourable electrostatic interactions between protein and RNA. However, at a sufficiently high ratio of RNA to protein, charge inversion leads to condensate dissociation and a reentrant phase transition to a homogenous, single-phase regime[42,46]. By varying the mixing ratio of G3BP1 and polyA RNA in a PhaseScan experiment, we observed reentrant phase behaviour of the resultant coacervate condensate (Fig. 2d). At a G3BP1 concentration of 1.8 µM, the system transitioned into and out of a phase-separated regime as the RNA concentration was varied between 7 and 40 ng/µL. This observation demonstrates the power of the PhaseScan technique, in accurately characterising reentrant phase transitions in a single experiment.

We then assayed the formation of phase-separated protein−protein coacervates by investigating the condensation of rotavirus proteins NSP2 and NSP5 (Fig. 2e). These proteins are key constituents of membraneless viral replication factories known as the viroplasm; recent studies suggest that LLPS provides a mechanism for viroplasm construction[47,48]. In agreement with these findings, and in support of a recent study demonstrating coacervation of NSP2 and NSP5 both in vivo and in vitro[12], we observe condensation of NSP2 and NSP5 in a concentration-dependent manner, with an NSP2-dependent threshold of NSP5 relative to NSP2 required for LLPS to occur.

Finally, we probed phase separation of the nucleocapsid protein of the human coronavirus SARS-CoV-2 (SARS-CoV-2 N) (Fig. 2f). In accordance with previous findings[49,50], we observe phase separation in an RNA-dependent manner. The ordinarily homogeneous SARS-CoV-2 N protein phase separated in the presence of polyA RNA concentrations below 25 ng/µL, however no phase separation was visible at RNA concentration above 65 ng/µL. The exact position of the phase boundary varied according to SARS-CoV-2 N concentration, with a higher protein concentration requiring a correspondingly higher concentration of RNA for phase separation to be ablated. These results

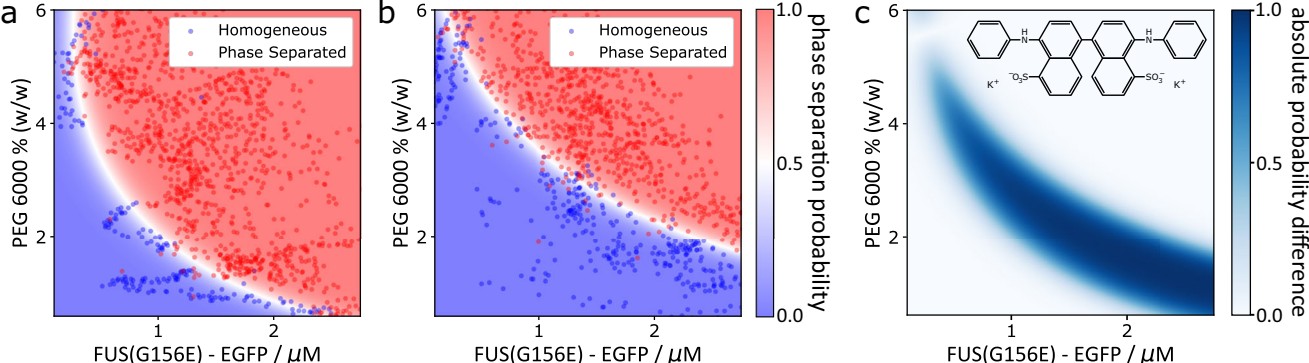

**Fig. 3 | Probing the effect of small molecule modulators on phase separation using PhaseScan. a** Phase diagram of EGFP-tagged EGFP-tagged FUS[G156E] vs. PEG 6000 concentration in the absence of bis-ANS. $N = 1616$ droplets. **b** Phase diagram of EGFP-tagged EGFP-tagged FUS[G156E] vs. PEG 6000 concentration in the presence of bis-ANS. $N = 2559$ droplets. **c** Molecular structure of bis-ANS and differential phase diagram of homotypic FUS phase diagram ± bis-ANS. Red and blue data points in scatter plots correspond to individual microdroplets classified as phase separated or homogeneous, respectively. The heat map corresponds to the probability of phase separation as determined by an SVM classifier trained on the droplet scatter plot. Source data are provided as a Source Data file.

are in agreement with previous studies, which have shown through manual experiments that phase separation of SARS-CoV-2 N can be driven by the presence of RNA, but that excess nucleic acid concentrations result in condensate dissolution[50].

In summary, these experiments demonstrate that the PhaseScan approach is applicable to a broad range of condensate systems. The assay can be used to characterise homo- and heterotypic phase separation of full-length proteins, including the observation of re-entrant phase separation behaviour in protein–RNA coacervates. PhaseScan can also characterise phase separation driven by protein–protein coacervates, as well as LLPS of short peptide sequences, which we demonstrate by examining the phase separation of the proline–arginine dipeptide (PR)₂₅ with polyU RNA (Supplementary Fig. 6).

## Observation of small-molecule modulation of phase separation

Next, we applied our platform to the investigation of the effect of small molecules on phase separation. There is great interest in identifying compounds that modulate LLPS as condensate forming processes are associated with a wide range of diseases, including neurodegenerative disorders and cancer[14,51–53]. Critical to this effort is the provision of high-resolution phase diagrams for the accurate quantification of the effect of small-molecule modulators on the phase separation equilibrium. Moreover, for high-throughput screening of drug candidates, it is imperative that the phase diagram for a particular drug–protein combination is acquired rapidly. The PhaseScan platform brings together these features and enables fast, automated generation of different solution conditions for the high-resolution assessment of changes in protein phase behaviour.

To establish the applicability of the PhaseScan system for probing the effects of small-molecule compounds, we first determined the modulation of FUS[G156E] phase separation by the small molecule 4,4′-dianilino-1,1′-binaphthyl-5,5′-disulfonic acid (bis-ANS), a compound which has recently been shown to act as a potent modulator of various phase separating proteins, including the FUS low-complexity domain[53]. To this end, we modified the microdroplet generator to allow the addition of a fourth solution component to the PhaseScan experiment (Supplementary Fig. 7), which in this case contained the small-molecule modulator bis-ANS. This solution was injected into the PhaseScan experiment at a constant flowrate, in addition to EGFP-tagged FUS[G156E], PEG and buffer solutions at varying flow rates, to achieve an equal concentration of the small molecule in each droplet (see Methods, Droplet Generation). Bis-ANS was at a concentration of 1 mM (in 1% DMSO) and was diluted 10-fold on chip to results in a final concentration of 100 μM (0.1% DMSO) in the

droplet. As a control, we performed an experiment without bis-ANS but with buffer supplemented with 1% DMSO (i.e., 0.1% DMSO in the droplet). As shown in Fig. 3a, b, the position of the phase boundary shifted markedly towards lower protein and lower PEG concentrations in the presence of bis-ANS, in line with previous observations on the LCD of FUS[53]. A differential map of the SVM-derived heat maps quantifies the shift of the phase boundary (Fig. 3c), thus providing a means to quantitatively assess the effect of chemical modulators on phase behaviour.

In summary, the PhaseScan approach provides a high-resolution assessment of changes in protein LLPS behaviour upon small-molecule addition. This feature is significant, as the effect of many candidate molecules is likely to be subtle, and identification of potential hits from a naïve candidate panel would require high assay resolution. PhaseScan achieves this rapidly with minimal sample consumption, promising efficient scale-up for high-content library screening.

## Generation of multi-dimensional phase diagrams

It has been established that a variety of environmental factors can trigger or modulate biomolecular phase separation. Investigation of these effects is typically constrained to analysis in two-dimensional (2D) chemical space, primarily due to the impractically large number of individual experiments that must be carried out if multiple parameters are investigated manually. Since the PhaseScan platform enables automated and high-throughput generation of a wide range of solution conditions, we sought to expand the platform to enable the investigation of multi-dimensional chemical space beyond conventional two-dimensional phase diagrams in a single experiment.

To demonstrate this, we performed a three-dimensional (3D) parameter space PhaseScan experiment on FUS[G156E]. We included 1,6-hexanediol (1,6-HD), a small molecule known to interfere with LLPS behaviour that has been employed extensively to probe condensate systems[19]. This was added as a fourth component to the PhaseScan mix in addition to EGFP-tagged FUS[G156E], PEG and buffer solutions (Supplementary Fig. 7). Following droplet generation, trapping and imaging (~7500 droplets), the relative concentrations of each of the three barcodes was used to determine the concentrations of FUS[G156E], PEG and 1,6-HD present in each drop. This measurement was combined with the classification of each droplet as containing phase separated or homogeneous protein to construct a three-dimensional phase diagram for the system (Fig. 4a, b, Supplementary Movie 1). As before, the concentration of EGFP-tagged FUS[G156E] and PEG were barcoded by the EGFP tag and the addition of Alexa647 dye, respectively, whereas the concentration of 1,6-HD was barcoded by the addition of Alexa546 (Fig. 4c).

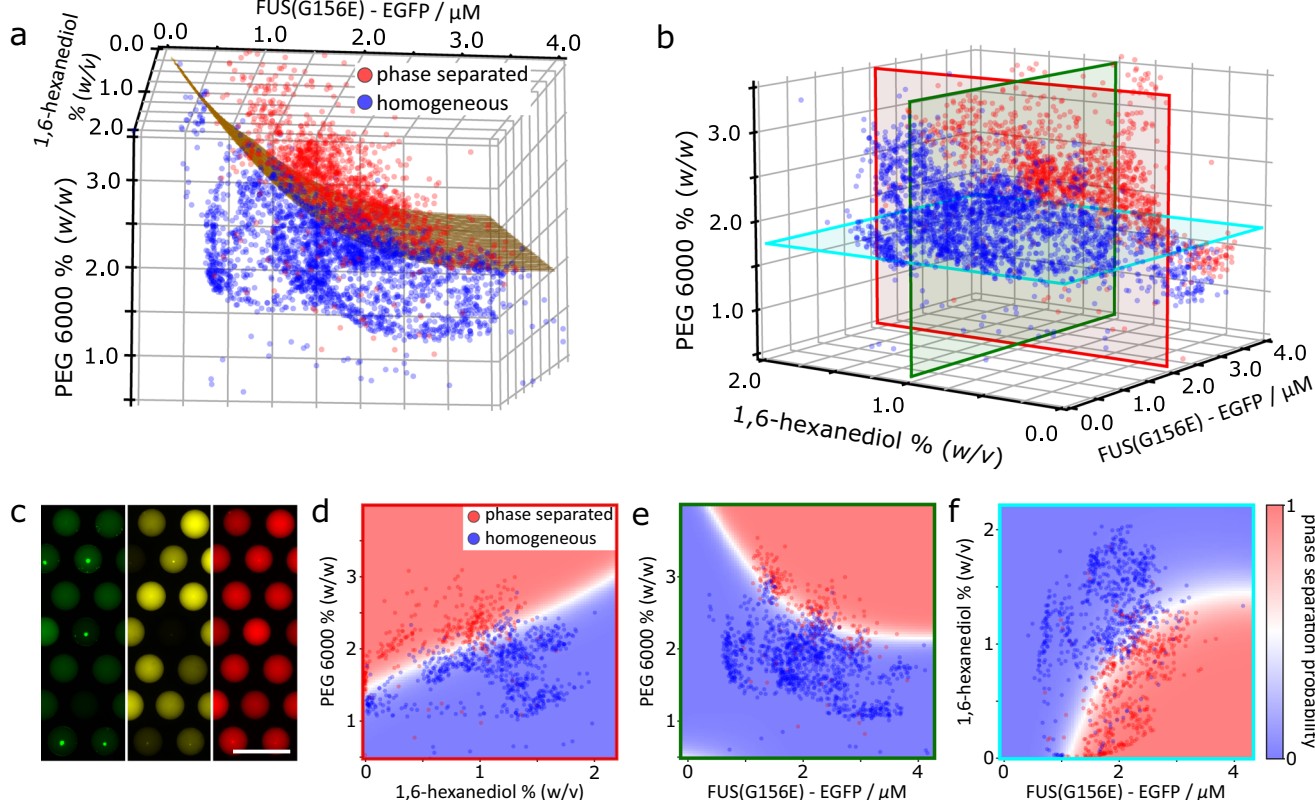

**Fig. 4 | Generation of multidimensional phase diagrams using the PhaseScan platform. a, b** 3D phase diagram of EGFP-tagged FUS$^{G156E}$ vs. PEG 6000 vs. 1,6-HD concentration. $N = 3904$ droplets. **c** Epifluorescence microscopy images of trapped microdroplets with EGFP (green), Alexa546 (yellow), and Alexa647 (red) fluorescence corresponding to FUS$^{G156E}$ and 1,6-HD, and PEG concentrations, respectively. **d–f** 2D slices of the 3D phase diagram, with outline colours corresponding to the red, green and blue planes shown in (b), respectively. Red and blue data points in scatter plots correspond to individual microdroplets classified as phase separated or homogeneous, respectively. The heat map corresponds to the probability of phase separation as determined by an SVM classifier trained on the droplet scatter plot. Source data are provided as a Source Data file.

As observed in the 2D phase diagrams shown previously, and evident in the projection from the 3D phase diagrams (Fig. 4d–f), phase separation was favoured at high protein and PEG concentrations. As expected, phase separation was reduced at higher concentrations of 1,6-HD, since 1,6-HD is known to disrupt protein condensates by competing for hydrophobic interactions that commonly drive protein LLPS. Notably, the antagonistic effects of molecular crowding and 1,6-HD on phase separation as a function of protein concentration are simultaneously observable (Fig. 4e–g). We propose that the ability to assess multiple modulators of phase separation in a single experiment presents a facile, high-resolution means with which to investigate mechanistic aspects of phase separation with minimal sample consumption (<6 μL of protein solution).

Biomolecular condensation has transformed our understanding of cell biology. Physicochemical characterisation of the parameters that control and modulate phase separation has thus become essential for an improved understanding of protein phase behaviour, including for the therapeutic modulation of LLPS phenomena. The PhaseScan platform presented herein provides a basis for the rapid and high-resolution acquisition of LLPS phase diagrams through the application of microdroplet techniques. We have demonstrated that the PhaseScan approach is applicable to a wide range of phase-separating systems ranging from coacervating peptides, human proteins and viral proteins, and provides a useful tool to enable rapid, high-resolution mechanistic analysis of phase separation processes. PhaseScan further enables the provision of high-resolution phase diagrams for the accurate quantification of the effect of small molecule modulators on the phase separation equilibrium, a feature that is essential in drug screening efforts. PhaseScan allows exploration of higher-dimensional chemical space in a single experiment, including the effect of small-molecule modulation, with minimal sample consumption.

We envisage that microfluidic platforms such as those presented here can find many applications in the quantitation of LLPS equilibria, such as, for example, the screening of candidate small molecules for the therapeutic modulation of phase-separation behaviour. Moreover, together with upstream integration of automated sample handling from microwell plates[54], rapid screening of multiple drug-protein combinations could be achieved. We note that with the relatively low droplet generation rate we employ here (~30 Hz), future experiments could achieve an improved assay throughput by combining high-frequency droplet generation with fast microscopy techniques. This approach opens a route towards high resolution and high throughput exploration of protein and nucleic acid phase behaviour under a rich variety of conditions.

## Methods

### Materials
All reagents and chemicals were purchased with the highest purity available. Tris as a buffer reagent and KCl were from Fisher Scientific. PolyU and PolyA RNA with a molecular weight range from 800–1000 kDa was purchased from Sigma Aldrich as lyophilized powder. PolyU$_{100}$ RNA was purchased from Biomers as lyophilized power and dissolved into a stock of 1 mg/mL in mQ water before use 1,6-HD was purchased from Santa Cruz Biotechnology. Bis-ANS was purchased from Invitrogen and dissolved to make a stock solution in DMSO (Sigma Aldrich). Alexa546 and Alexa647 carboxylic acid were

obtained from Thermo Fisher. PEG 4000 and 6000 were from Sigma Aldrich. Details on the production of proteins (FUS[G156E]-EGFP, G3BP1-GFP, NSP2/5, SARS-CoV-2 N and (PR)$_{25}$) are given in the Supplementary Information.

## Droplet generation

Syringe pumps (neMESYS modules, Cetoni) were used to control flows of protein, buffer and phase separation trigger solutions to the microfluidic device. The syringe (Hamilton 1710) and tubing (PTFE, 0.012"ID × 0.030"OD, Cole-Parmer) for the protein sample were pre-filled with FC-40 oil, before the small working volume of protein sample (10–20 µL) was aspirated into the tubing. 3 µM Alexa647 and Alexa 546 carboxylic acid solutions were pre-mixed into the solutions of phase separation trigger and phase separation modulator (if used), respectively, as fluorescence barcodes. The aqueous flowrates were configured to vary automatically according to a pre-programmed flow profile, with total aqueous flow of 60 µL/h, to produce the desired concentration of droplet components. FC-40 oil containing 1% w/v fluorosurfactant (RAN Biotechnologies) was introduced to the device at a constant flow of 100 µL/h. Generated droplets were transferred via tubing to a separate microfluidic device where they were trapped in a floating droplet array[38]. After sufficient droplet generation time (5 min), the aqueous flow was turned off and oil flow was used to flush excess drops out of the array device. In some experiments, several droplet-catcher devices were linked in sequence to trap the desired number of droplets.

## Protein production

FUS[G156E]-EGFP expression and purification was adapted from Patel et al[9]. In short, recombinant protein production was performed in Sf9 insect cells (Expression Systems, Cat#94-001 F) using the baculovirus system[55]. The protein was produced as a C-terminal EGFP fusion with an N-terminal maltose-binding protein (MBP) tag and a C-terminal hexahistidine (His$_6$) tag. Cells expressing MBP-FUS-EGFP[G156E]-His$_6$ were harvested 72 h post-infection, centrifuged, and then resuspended in 50 mM Tris-HCl (pH 7.4), 1 M KCl, 5% (w/v) glycerol, 1 mM DTT, 10 mM imidazole supplemented with EDTA-free protease inhibitor cocktail set III (Calbiochem) and 0.25 U/mL benzonase (provided by the protein expression facility of the Max Planck Institute of Molecular Cell Biology and Genetics (MPI-CBG), Dresden). After cell lysis, using a shear homogenizer (Microfluidics), the protein was purified by immobilized-metal affinity chromatography (IMAC) using nickel-nitrolotriacetic acid (Ni-NTA) columns (Macherey-Nagel). The column was washed with lysis buffer. Elution was done with 50 mM Tris-HCl (pH 7.4), 1 M KCl, 5% (w/v) glycerol, and 500 mM imidazole. His$_6$ and MBP tags were proteolytically removed with 3C-His$_6$ preScission protease (provided by the protein expression facility of the MPI-CBG, Dresden). Protein was further purified by size-exclusion chromatography (SEC) using a Superdex 200 pg 26/600 column (GE Healthcare). Aliquots containing the protein were flash-frozen and stored at −80 °C. The protein was stored in 50 mM Tris-HCl (pH 7.4), 500 mM KCl, 1 mM dithiothreitol (DTT), 5% (w/v) glycerol.

G3BP1-GFP expression and purification was carried out as described in Guillén-Boixet et al.[45]. Briefly, recombinant His$_6$-GFP-G3BP1-MBP was expressed in and purified from insect cells (Expression Systems, Cat#94-001 F) using a baculovirus expression system[55]. Following lysis (EmulsiFlex-C5, Avestin) in buffer containing 50 mM Tris-HCl (pH 7.5), 1 M KCl, 2 mM EDTA, 2 mM DTT and 1x EDTA-containing protease inhibitor cocktail (Roche), the protein was purified by affinity chromatography using amylose resin (New England Biolabs) to capture the protein via its MBP tag from the supernatant of the cell lysate. The sample was then subjected to IMAC using Ni-NTA resin (Qiagen). The column was washed with an EDTA-free lysis buffer containing 20 mM imidazole. The protein was subsequently eluted from the Ni-NTA column with 250 mM imidazole. His$_6$ and MBP tags were cleaved

off with PreScission protease during an overnight dialysis step at 4 °C. The protein was further purified by SEC using a HiLoad 16/600 Superdex 200 pg (GE Healthcare) on an Akta Ettan system in 50 mM Tris-HCl (pH 7.5), 300 mM KCl, 1 mM DTT buffer. Aliquots containing the protein were flash-frozen and stored at −80 °C.

NSP2 from Rotavirus A (strain RF) was produced as a C-terminal His$_6$-tagged protein (NSP2-His$_6$) in BL21(DE3) Escherichia coli transformed with a pET-28b-NSP2 construct, as previously described[56,57]. Expression was carried out at 24 °C in Luria Bertani (LB) media supplemented with 1% (v/v) glucose, and cultures were induced with 0.5 mM IPTG once they reached optical density (OD$_{600}$) of 0.6, and harvested 14 h post-induction. Cell pellets were resuspended in 50 mM Tris-HCl (pH 8), 300 mM NaCl, supplemented with 100 µg/mL chicken egg lysozyme (Sigma), 0,5% Tween 20 and a complete protease inhibitor (Roche), followed by DNAseI treatment (10 µg/mL, Roche) for 15 min prior to clarifying the lysate by centrifugation for 20 min at 15,000 × g at 4 °C. Clarified lysate was loaded on a 5 mL His-Trap HP column (GE Healthcare), followed by a wash step and elution with 0.5 M imidazole, 20 mM HEPES-Na (pH 7.5). Eluted peak fractions were pooled, diluted with 10 mM HEPES-Na (pH 7.5) and purified by ion-exchange chromatography (IEX) step using a CaptoImpRes SP column (GE Healthcare). Eluted peak fractions were further resolved by SEC on a Superdex 200 10 × 300 GL column (GE Healthcare) pre-equilibrated with 25 mM HEPES-Na, pH 7.5, 150 mM NaCl. Purified protein aliquots were snap-frozen and stored at −80 °C for subsequent use. Labelling of the purified His-tagged NSP2 protein was achieved by pre-incubating 10 µM NSP2 with 1 µM Atto488-NTA dye (Sigma) for 5 min. Following incubation, the protein was immediately used.

Recombinant NSP5 from rotavirus A (strain RF) was produced as a N-terminal Strep-tagged protein in BL21(DE3) E.coli transformed with a pET-28b-NSP5 construct, as described[12]. Cultures (LB medium as above with NSP2) were induced with 1 mM IPTG, and expression was carried out for 6 h at 37 °C. After enzymatic lysis as described above for NSP2, the protein was purified under denaturing conditions, followed by its refolding, as described[58]. Briefly, washed inclusion bodies were solubilized in 6 M guanidinium hydrochloride and the protein-containing fraction was then subjected to a refolding protocol following step-wise dialysis. After refolding, NSP5-containing fractions were further purified by IEX using a CaptoQ ImpRes column (GE Healthcare). Concentrated peak fractions were further resolved using SEC on a Superdex 200 10 × 300 column (GE Healthcare). Purified protein fractions were pooled, aliquoted and snap-frozen using liquid nitrogen, and stored at −80 °C for subsequent use.

For SARS-CoV-2 Nucleocapsid (N) protein purification, baculoviruses were produced from a pOCC102-His$_6$-MBP-3C-Nucleocapsid construct, leveraging the FlexiBAC approach as previously described. The N coding sequence was derived from SARS-CoV-2 lineage B (GenBank: MN908947.3)[59] and subsequently codon-optimized for insect cell expression. Baculoviruses were then used in conjunction with an in-house Sf9 insect cell expression system (provided by the protein expression facility, MPI-CBG)[55]. Briefly, 0.5 L Sf9 cells (-10$^6$ cells mL$^{-1}$) were infected with 2% (v/v) baculoviral supernatant and subsequently grown for 72 h at 27 °C and 85 rpm. Cell pellets were obtained by centrifugation at 300 × g for 5 min and routinely flash frozen in liquid nitrogen, and then stored at −80 °C. For cell lysis, Sf9 pellets were thawed and resuspended in 50 mL cold lysis buffer (1 M NaCl, 50 mM Na$_x$H$_x$PO$_4$, 20 mM imidazole, 5% (v/v) glycerol, 4 mM MgCl$_2$, 1 mM DTT, 1× complete protease inhibitor, 1 U mL$^{-1}$ DNase I, pH 7.4) and passed through an LM20 microfluidizer (15,000 psi, 4 °C). Following ultracentrifugation (70,000 × g, 4 °C, 30 min), the obtained supernatant was passed through a 0.45 µm filter and subjected to a three-step FPLC purification on an ÄKTA pure 25 M chromatography system (Cytiva) at room temperature. Filtered cell lysates were first passed, at 5 mL min$^{-1}$, through the Ni$^{2+}$ NTA resin of a preequilibrated 5 mL HisTrap HP column (Cytiva). This was followed by a washing step

with 40 mL imidazole wash buffer (150 mM NaCl, 50 mM $Na_xH_xPO_4$, 20 mM imidazole, 5% ($v/v$) glycerol, pH 7.4), and an imidazole gradient elution over 50 mL, reaching 300 mM imidazole, under otherwise identical buffer conditions. Fractions, of 1.5 mL each, were collected throughout and those with high absorbance intensities at 280 nm were analysed by SDS polyacrylamide electrophoresis and subsequently pooled. Following the concentration of pooled fractions, supplemented with additional 400 mM NaCl, 150 mM Arg-HCl (pH 7.4) and 300 mM trehalose to prevent aggregation, by repeated centrifugation ($4000 \times g$, 25 °C, 3 min) in a 30 kDa cut-off filter column to ~5 mL, purified proteins were digested using 500 µg of 3 C protease (in-house purification; protein purification facility, MPI-CBG) in the presence of 0.5 mM DTT over 1 h at 25 °C to remove the MBP solubility tag. Upon dilution of the reaction mixture to a final NaCl concentration of ~150 mM, the proteins were passed through the heparin-conjugated resin of a preequilibrated 5 mL HiTrap Heparin HP column (Cytiva). This was followed by a washing step with 40 mL heparin wash buffer (150 mM NaCl, 50 mM $Na_xH_xPO_4$, 5% ($v/v$) glycerol, pH 7.4) and a high salt gradient elution over 50 mL, ultimately reaching 1 M NaCl and pH 7.4, under otherwise similar buffer conditions. Fractions with putatively high protein content were again analysed and subsequently pooled. The pooled fractions were supplemented with additional 700 mM NaCl, 150 mM Arg-HCl (pH 7.4) and 300 mM trehalose to prevent aggregation, and concentrated by repeated centrifugation ($4000 \times g$, 25 °C, 3 min) in a 30 kDa cut-off filter column to ~2 mL. The concentrate was then passed through a 0.2 µm spin filter and resolved by SEC at a reduced flow rate of 0.5 mL min$^{-1}$ on a preequilibrated Superdex 200 Increase 10/300 GL column (~24 mL column volume; Cytiva), in SEC buffer (50 mM $Na_xH_xPO_4$, 300 mM NaCl, 5% ($v/v$) glycerol, 1 mM DTT, pH 7.4), which also constituted the final storage buffer. Following SEC, analysed and pooled fractions were again concentrated, as described above, in this case to ~200 µL. Employing a ND-1000 spectrophotometer (Thermo Scientific) the protein concentration was then determined at 280 nm (with a molar extinction coefficient ε ~43,900 M$^{-1}$ cm$^{-1}$ of the ~46 kDa CoV-2 N protein) and the extent of nucleic acid contamination evaluated based on the 260 to 280 nm absorption ratio (obtaining a value of ~0.56; a ratio of ≥0.7 indicating nucleic acid contamination). Subsequently, 5 µL aliquots were prepared, flash frozen in liquid nitrogen, and stored at −80 °C.

The (PR)$_{25}$ peptide, containing 25 proline–arginine repeats, was obtained from GenScript. N-terminally labelled PR25 was obtained by reacting the peptide with amine-reactive AlexaFluor546 (Sigma Aldrich). (PR)$_{25}$ experiments were conducted using a mix of 10% labelled and 90% unlabelled peptide.

## Device design and fabrication

PDMS (Corning) devices for droplet generation and multilayer well-devices for droplet collection and imaging were produced on SU-8 (Microchem) moulds fabricated via photolithographic processes as described previously[60–62]. The design of the device was drawn with AutoCAD (AutoDesk) and then printed on a photomask (Micro Lithography). The pattern of the mask was transferred by UV exposure[63] to a polished silicon wafer coated with a 50 µm thick layer of SU8-3050 photoresist (Microchem). Excessive SU-8 photoresist was removed using propylene glycol methyl ether acetate (PGMEA; Sigma). The wafer with SU-8 patterns (i.e., master) was dried by blowing with nitrogen and baking at 95 °C. The master was placed in a plastic petri dish and served as a mould for poly(dimethylsiloxane) (PDMS; Sylgard184, Dow Corning) casting. The PDMS base and crosslinking agent was mixed in a 10:1 ratio and polymerised by baking at 60 °C for 2 h. The PDMS device was cut from the petri dish with a scalpel and cleaned by sonication in an ethanol bath for 15 min. Both the PDMS replica and the glass slide were activated in an oxygen plasma oven (30 s, 40% power, Femto, Diener Electronics) before

bonding. The channels were treated with 1% ($v/v$) tri-chloro(1H,1H,2H,2H-perfluorooctyl)silane (Sigma) in HFE-7500 (Fluorochem) for 1 min, before being dried with nitrogen and heated on a hotplate at 95 °C for 10 min.

## Imaging

Trapped microdroplets were imaged using an AxioObserver D1 microscope (Zeiss) equipped with a 5× air objective (Zeiss) and a high-sensitivity camera (Evolve 512, Photometrics, Metamorph 6.3 image acquisition software), except for the data in Fig. 2d which was acquired by imaging droplets under flow, using a microscope equipped with a dichroic filter set for simultaneous multi-wavelength imaging. Appropriate filter sets were used for EFGP, Alexa Fluor 546 and Alexa Fluor 647 detection (Chroma Technology 49002, 49004 and 49006, respectively). Minimal crosstalk between fluorescence channels was observed, which was removed during the image processing and calibration procedure (see Supplementary Note 6 and Supplementary Fig. 9).

## Droplet detection and phase diagram generation

Acquired images were analysed using a custom-written Python script (Python version 3.8, SVM algorithms are a component of the scikit-learn Python library). Representative data and additional details are provided in the Supplementary Information (Supplementary Figs. 9 and 10). Briefly, droplets were fitted as circles in the images. Non-circular droplets or erroneous detections were filtered and removed. From the fitted circular areas, the total intensity was calculated and normalised to obtain the intensity per unit volume (calculated using the fitted diameter), and converted to concentrations by comparison to calibration images acquired with known barcode concentration (see Supplementary Note 6 and Supplementary Fig. 9). Droplets were classified as phase-separated or homogeneous according to the presence or absence of at least three connected pixels of an intensity above an algorithm-defined threshold intensity. Representative classification output is presented in Supplementary Fig. 10. Droplet classification and barcode concentrations were then combined on a per-droplet basis to produce phase diagrams. A Support Vector Machine algorithm was then trained on the droplet dataset to produce a probability map of phase separation over the phase-space in question.

## Statistics and reproducibility

Each phase diagram was constructed with a sample size of >1500 droplets. From bootstrapping analysis, this sample size resulted in a difference in SVM phase diagram generation of <3% between sampling repeats (see Supplementary Note 4 and Supplementary Fig. 4). During image analysis of trapped microdroplets, data were excluded for overlapping or touching droplets which prevented quantitative droplet analysis. Examples of this process are shown in Supplementary Fig. 10. Investigators were not blinded to allocation during experiments and outcome assessment.

## Reporting summary

Further information on research design is available in the Nature Portfolio Reporting Summary linked to this article.

# Data availability

Source data are provided with this paper and are available in the FigShare repository under accession code 10.6084/m9.figshare.21405129.

# Code availability

Custom Python scripts developed for the study are available at GitHub https://github.com/rqi14/PhaseScan

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

## Acknowledgements

The research leading to these results has received funding from the EPSRC (award RG90413, W.E.A.), the Royall Scholarship (N.A.E.), a Krishnan-Ang Studentship (R.Q.), Trinity College (Cambridge Honorary Trinity-Henry Barlow Scholarship; R.Q.), the Cambridge Trust (Honorary International Scholarship; R.Q.), the European Research Council under the European Union's Horizon 2020 Framework Programme through the Marie Skłodowska-Curie grant MicroSPARK (agreement no. 841466; G.K.), the Herchel Smith Fund of the University of Cambridge (G.K.), the Wolfson College Junior Research Fellowship (G.K.), the Winston Churchill Foundation of the United States (T.J.W.), the Harding Distinguished Postgraduate Scholar Programme (T.J.W.), the Wellcome Trust Henry Wellcome fellowship 218651/Z/19/Z (J.N.A), the Wellcome Trust [213437/Z/18/Z] (A.B.), the Canadian Institutes of Health Research (Foundation Grant and Canadian Consortium on Neurodegeneration in Aging Grant) (P.StGH.), US Alzheimer Society Zenith Grant ZEN-18-529769 (P.St.G.-H.), Alzheimer Society of Ontario Chair in Alzheimer's Disease Research (P.St.G.-H.), a Wellcome Trust Collaborative Award 203249/Z/16/Z (P.St.G.-H., T.P.J.K.), the European Research Council under the European Union's Seventh Framework Programme (FP7/2007-2013) through the ERC grant PhysProt (agreement no. 337969; T.P.J.K.) and the Frances and Augustus Newman Foundation (T.P.J.K.).

## Author contributions

W.E.A., G.K. and T.P.J.K. conceived the study. W.E.A., N.A.E., R.Q., K.D., T.J.W., J.A., J. N.-A. and Y.X. performed experiments. R.Q. and W.E.A. developed analytical tools. T.M.F., J.G.-B., A.B., D.K., S.A. and P.St.G.-H. provided materials. W.E.A. and G.K. wrote the original draft, all authors reviewed and edited the paper.

## Competing interests

Parts of this work pertaining to the encapsulation and analysis of phase-separating macromolecules within microdroplets have been the subject of a patent publication (application number GB2021051243W, publication number WO2021234410A1) filed by Cambridge Enterprise Limited, a fully owned subsidiary of the University of Cambridge, W.E.A., G.K., T.J.W., R.Q. and T.P.J.K. are inventors. T.P.J.K. and P.St.G.-H. are founders, and W.E.A., G.K. T.J.W. and R.Q. are consultants and/or employees of Transition Bio Ltd. The remaining authors declare no competing interests.
