## [Peer Review File · Nature Communications]

Reviewers' Comments:

Reviewer #1:

Remarks to the Author:

The current manuscript by Knowles and Colleagues describes a microfluidic platform to generate in vitro protein or protein-RNA mixture phase diagram. This lab-on-a-chip termed as PhaseScan, was used to get phase separation diagrams for FUS and other condensate systems. The data are clearly presented and mostly serve as proof of principle since the systems studied were previously known at some length including the effect of Bis-ANS. Overall, the manuscript is easy to follow. My comments are below:

- a) I am slightly confused about what is the focus of this manuscript. If it is technology development, I am wondering how novel this is since similar technologies were discussed in the literature over a decade ago (for example, see work by Freden group describing "phasechip" <https://pubs.rsc.org/en/content/articlelanding/2010/lc/b925661j>). More recently, other groups have applied microfluidics to study the phase behavior of biomolecular condensates as well. See <https://pubs.acs.org/doi/abs/10.1021/acs.analchem.1c02700>. There are other examples as well. The phasechip technology allowed a temperature phase diagram, which is more advanced than what is presented in the current manuscript. These works were not cited or discussed, making it difficult for one to judge how novel this current technology is.
- b) Since the data do not really provide any new insights regarding the behavior of the systems studied, can the technology be extended beyond simple proof of principle data collection, thus strengthening the manuscript with newer findings?
- c) I beg to differ from the authors in that they are NOT measuring phase diagrams. These are not phase diagrams, which have a very specific meaning and signify the measurement of binodals. These are state diagrams at best.
- d) The data in Fig 1i and its interpretation are clear. This is not the case for data shown in Fig 2f, 3b, and Fig 3d-f. They are much more scattered and the designation of the probability and a clear demarcation of the "phase boundary" may be problematic. Can the data quality be improved?
- e) Concentration calibration through an epi-fluorescence microscope has limitations. Can this be done independently in a confocal setup to verify accuracy?
- f) The word "phase space" has a specific meaning in classical mechanics (https://en.wikipedia.org/wiki/Phase_space) please avoid such terminologies.
- g) Minor point: Please share the image analysis codes for public access if this is not done already.

Reviewer #2:

Remarks to the Author:

Cellular phase separation resulting in biomolecular condensates has in recent years increasingly emerged as an important contributor to biological function and disease. In vitro studies of this process with components of these cellular condensates have provided very important insights and models that can be used to interpret and predict cellular behavior and control. An important aspect of many such studies is the measurement of phase behavior of the system as a function of variables such as component concentrations, ionic strength and small molecule modulators. However, such studies are often laborious and sample consuming, and thus limited in terms of the variable space explored. Along these lines, microfluidic methods are particularly well suited to exploring a large parameter space, while also using small amounts of precious samples. The authors here describe such a much-improved method that builds on previous droplet microfluidic methodology that some of the authors have reported. The results are impressive. Using their microfluidic technology along with in-situ barcoded readouts of parameter values and further analysis, they are able to scan a large array of droplets/conditions for several protein systems, and

to demonstrate modulation by solution conditions (including crowding and ionic strength), RNA and small molecules. In addition, the high throughput allows multidimensional analyses to be carried out, which is even more difficult using conventional methods. These strengths are combined with low sample consumption (few microliters even for the multidimensional phase diagram) and fast data acquisition, together expected to be very useful for both fundamental studies as well as potentially in drug discovery efforts. Overall, the work is very well done, and the method will be valuable for a broad range of systems in this rapidly expanding scientific area. Therefore, I believe that the work will be of substantial interest and utility to the scientific community. I have noted below some minor comments/suggestions that the authors should consider/address.

It would be useful for the authors to include information about any measured/estimated false positive probabilities of droplet classification (into well-mixed vs. phase separated) using their analysis.

In a related point, the measured barcode concentration errors seem significant (Fig 8h). In future work, could this measurement precision (and the corresponding errors in the phase diagrams) be improved by increasing the barcode dye concentration, or are the authors concerned about potentially perturbing the phase separation process?

I suggest that the authors include Reference 35 in the list/discussion in the 1st paragraph on page 4.

Since this is a new method for measuring phase diagrams, it would be useful to include in the main text data for a comparison with diagrams measured using more traditional methods. Hence, I suggest including the data from Fig S2 in Fig 1i.

Reviewer #3:

Remarks to the Author:

Description of the full phase diagram of a phase-separating (protein-RNA) system is a challenge in a multi-dimensional space of many parameters, which requires a combinatorial approach enabling many parallel experiments. In the paper, the authors present a combinatorial droplet microfluidic platform (PhaseScan) that enables the high-resolution acquisition of LLPS phase diagrams in multiple chemical and physical dimensions. The microfluidic platform enables fast and cost-effective generation of many droplets of various composition, and barcoding of components enables to generate multidimensional, detailed phase diagrams. The platform provides a basis for the rapid and high-resolution acquisition of LLPS-related data, as demonstrated on a variety of phase-separating systems, ranging from coacervating peptides, human proteins and viral proteins, including even characterizing the effect of small-molecules. There are some minor issues, which should be addressed for improving the manuscript, before acceptance can be considered.

Minor comments

- 1) It should be demonstrated on at least one system how sensitive the system is to a broad range of protein concentrations?
- 2) On page 3. when mentioning that phase separation has been established for hundreds of proteins, it should be mentioned that the issue of what a "phase separating" protein is, is not trivial, as outlined, for example, in Farahi et al. (2021) Int J Mol Sci. 22: 3017. In this paper, different databases have been shown to present very different collections of phase-separating proteins, outlining the underlying cause that proteins may have very different roles in LLPS.
- 3) FUS is studied in several different contexts, it is not clear why a particular mutant is used. Maybe it is related to disease, and its comparison to WT would have been very instructive in terms of disease mechanism?
- 4) A deep underlying issue of these studies is that phase separation is not an equilibrium phenomenon, LLPS droplets are nucleated and evolve by Ostwald ripening. Therefore, observations by the PhaseScan system may depend on delay times typical for the system. This should be at least mentioned and discussed in the paper: maybe microfluidics enables to approach kinetic aspects of phase separation?

5) It is central to the LLPS field that charge compensation by RNA often results in "reentrant" behaviour. The power of the system could be demonstrated if this was apparent (e.g., on Figure 2a, e or f).

6) when the effect of a small molecule (bis-ANS) is demonstrated, it would be very interesting to show that the technique enables screening of multiple conditions (compound concentration, or different compounds).

7) In conclusion, whereas phase diagrams of many data points are presented, one misses the point why this technique is superior to previous systems: a novel insight in at least one system, not apparent in previous lower-resolution approaches, should be demonstrated.

Response to the Reviewer's Comments

Reviewer #1:

The current manuscript by Knowles and Colleagues describes a microfluidic platform to generate in vitro protein or protein-RNA mixture phase diagram. This lab-on-a-chip termed as PhaseScan, was used to get phase separation diagrams for FUS and other condensate systems. The data are clearly presented and mostly serve as proof of principle since the systems studied were previously known at some length including the effect of Bis-ANS. Overall, the manuscript is easy to follow. My comments are below:

Reply: We thank the reviewer for the careful assessment of our work and the positive feedback. Below we provide a point-by-point response to the reviewer's valuable comments.

a) I am slightly confused about what is the focus of this manuscript. If it is technology development, I am wondering how novel this is since similar technologies were discussed in the literature over a decade ago (for example, see work by Freden group describing "phasechip" <https://pubs.rsc.org/en/content/articlelanding/2010/lc/b925661j>). More recently, other groups have applied microfluidics to study the phase behavior of biomolecular condensates as well. See <https://pubs.acs.org/doi/abs/10.1021/acs.analchem.1c02700>. There are other examples as well. The phasechip technology allowed a temperature phase diagram, which is more advanced than what is presented in the current manuscript. These works were not cited or discussed, making it difficult for one to judge how novel this current technology is.

Reply: We thank the reviewer for raising this point. The focus of our manuscript is indeed the demonstration of the new PhaseScan technology. The key challenge in this space that we sought to address is access to phase diagrams with high data density (thousands of data points) distributed along arbitrary dimensions in chemical space. Non-droplet-based approaches (Li et al., 2022; Selimović et al., 2010) are not readily scalable to this level of data density, and existing droplet platforms which generate high data density (Bremer et al., 2020; Kopp et al., 2020) do so through shrinking droplets and thus have chemical dimensions which interdependent and cannot be controlled as independent variables. This activity highlights this space as an active area with significant interest, and we believe that the ability to acquire phase diagrams at high throughput with independent dimensions, as demonstrated in our paper, will unlock further significant possibilities. We have further expanded the set of references to include the ones highlighted by the referee.

b) Since the data do not really provide any new insights regarding the behavior of the systems studied, can the technology be extended beyond simple proof of principle data collection, thus strengthening the manuscript with newer findings?

Reply: We thank the reviewer for this helpful question. The primary focus of our manuscript is to communicate the methodology, generalisability, scalability, and potential applications of the PhaseScan assay. Specifically, we demonstrate for the first time to our knowledge a three-dimensional condensate phase diagram and, more generally, that a wide range of parameter space can be assessed in a practically feasible way by achieving multi-dimensional analysis of protein phase behaviour within a single experiment.

c) I beg to differ from the authors in that they are NOT measuring phase diagrams. These are not phase diagrams, which have a very specific meaning and signify the measurement of binodals. These are state diagrams at best.

Reply: This is an excellent point and one which we have thought about. The term 'phase diagram' is commonly applied to data such as is presented in the manuscript, whether generated microfluidically (Bremer et al., 2020; Li et al., 2022) or by conventional pipetting experiments (Guillén-Boixet et al., 2020; Yang et al., 2020), and we note that the other reviewers of this manuscript have themselves used

this terminology to describe our data. We have revised the manuscript to highlight that our measurements record the equilibrium position of the phase boundaries by showing that time-dependent measurements of the same condensate system produce identical phase diagrams (Figure S4).

d) The data in Fig 1i and its interpretation are clear. This is not the case for data shown in Fig 2f, 3b, and Fig 3d-f. They are much more scattered and the designation of the probability and a clear demarcation of the “phase boundary” may be problematic. Can the data quality be improved?

Reply: We thank the reviewer for raising this point. As shown in Figures 1, S4 and S9, our droplet analysis algorithm correctly classifies the vast majority of droplets in the data as homogeneous or phase separated. The large number of data points acquired by the PhaseScan approach means that the relatively small number of mis-classified drops do not significantly affect the determined position of the phase boundary.

However, we have chosen to replace the data for phase separation of PR25 with a newly acquired phase diagram for heterotypic G3BP1/RNA phase separation (Fig 2(d)) which has better imaging signal/noise and hence improved data quality relative to the PR25 example, which has been moved to the SI. The G3BP1 example demonstrates the applicability of PhaseScan for investigating reentrant phase-separating systems, with this phase diagram displaying reentrant phase separation behaviour as the system transitions from homogeneous to phase separated and back as the ratio of RNA to protein is increased at a given protein concentration.

e) Concentration calibration through an epi-fluorescence microscope has limitations. Can this be done independently in a confocal setup to verify accuracy?

Reply: Following the referee’s comment, we have detailed the calibration and reference imaging steps that were undertaken to achieve concentration measurements via epi-fluorescence microscopy in the SI and provided control data for determining barcode concentrations. In these measurements, barcode dye concentrations were first measured independently by UV-vis spectroscopy, these concentrations were then accurately recapitulated by our imaging and analysis protocol.

f) The word “phase space” has a specific meaning in classical mechanics (https://en.wikipedia.org/wiki/Phase_space) please avoid such terminologies.

Reply: We have followed the referee’s suggestion.

g) Minor point: Please share the image analysis codes for public access if this is not done already.

Reply: We agree, the analysis code will be made public on Github with publication.

Reviewer #2:

Cellular phase separation resulting in biomolecular condensates has in recent years increasingly emerged as an important contributor to biological function and disease. In vitro studies of this process with components of these cellular condensates have provided very important insights and models that can be used to interpret and predict cellular behavior and control. An important aspect of many such studies is the measurement of phase behavior of the system as a function of variables such as component concentrations, ionic strength and small molecule modulators. However, such studies are often laborious and sample consuming, and thus limited in terms of the variable space explored. Along these lines, microfluidic methods are particularly well suited to exploring a large parameter space, while also using small amounts of precious samples. The authors here describe such a much-improved method that builds on previous droplet microfluidic methodology that some of the authors have reported. The results are impressive. Using their microfluidic technology along with in-situ barcoded readouts of

parameter values and further analysis, they are able to scan a large array of droplets/conditions for several protein systems, and to demonstrate modulation by solution conditions (including crowding and ionic strength), RNA and small molecules. In addition, the high throughput allows multidimensional analyses to be carried out, which is even more difficult using conventional methods. These strengths are combined with low sample consumption (few microliters even for the multidimensional phase diagram) and fast data acquisition, together expected to be very useful for both fundamental studies as well as potentially in drug discovery efforts. Overall, the work is very well done, and the method will be valuable for a broad range of systems in this rapidly expanding scientific area. Therefore, I believe that the work will be of substantial interest and utility to the scientific community. I have noted below some minor comments/suggestions that the authors should consider/address.

Reply: We are delighted about the enthusiastic feedback on the approach, and we thank the referee for their helpful suggestions.

It would be useful for the authors to include information about any measured/estimated false positive probabilities of droplet classification (into well-mixed vs. phase separated) using their analysis.

Reply: We thank the reviewer for raising this important point. Droplet classification error is affected by the signal/noise of condensate fluorescence detection, and therefore varies between protein systems. Typically, the sum of false-positive and false-negative classification error is < 4% of the total number of droplets classified. We have included this information in the revised manuscript.

In a related point, the measured barcode concentration errors seem significant (Fig 8h). In future work, could this measurement precision (and the corresponding errors in the phase diagrams) be improved by increasing the barcode dye concentration, or are the authors concerned about potentially perturbing the phase separation process?

Reply: We agree with the reviewer that barcoding dyes may interfere with the phase separation process at higher concentrations, although we are confident that this is not the case for those we use in the manuscript (see Figure S3). Measurement precision can be enhanced in future work by improving the quality of the microscopy hardware employed (higher NA objectives, brighter illumination and higher detection sensitivity etc). However, an advantage of the high throughput inherent in the PhaseScan experiment is that measurements errors are averaged over many individual data points, allowing accurate determination of the phase boundary given sufficient droplets are assayed.

I suggest that the authors include Reference 35 in the list/discussion in the 1st paragraph on page 4.

Reply: We thank the reviewer for this suggestion. We have altered the position of this reference in the manuscript.

Since this is a new method for measuring phase diagrams, it would be useful to include in the main text data for a comparison with diagrams measured using more traditional methods. Hence, I suggest including the data from Fig S2 in Fig 1i.

Reply: We thank the reviewer for this comment and agree that a more direct comparison will aid interpretation of our results. We have combined Fig. S2 into Fig 1.

Reviewer #3:

Description of the full phase diagram of a phase-separating (protein-RNA) system is a challenge in a multi-dimensional space of many parameters, which requires a combinatorial approach enabling many parallel experiments. In the paper, the authors present a combinatorial droplet microfluidic platform (PhaseScan) that enables the high-resolution acquisition of LLPS phase diagrams in multiple chemical and physical dimensions. The microfluidic platform enables fast and cost-effective generation of many

droplets of various composition, and barcoding of components enables to generate multidimensional, detailed phase diagrams. The platform provides a basis for the rapid and high-resolution acquisition of LLPS-related data, as demonstrated on a variety of phase-separating systems, ranging from coacervating peptides, human proteins and viral proteins, including even characterizing the effect of small-molecules. There are some minor issues, which should be addressed for improving the manuscript, before acceptance can be considered.

Reply: We are delighted about the careful assessment our work and the positive feedback on the approach.

Minor comments

1) It should be demonstrated on at least one system how sensitive the system is to a broad range of protein concentrations?

Reply: We appreciate that applicability to a broad range of concentrations is important for the generalisability of our technique. In the data presented, protein concentration varies over two orders of magnitude (<1 μM – >20 μM). We have utilised protein in this concentration range since these concentrations are those relevant for the in-vitro phase separation of the proteins in question. In principle, the assay is applicable to any concentration of soluble protein, with the minimum concentration constrained by the sensitivity of fluorescence detection, which is primarily determined by the microscopy tools employed.

2) On page 3. when mentioning that phase separation has been established for hundreds of proteins, it should be mentioned that the issue of what a “phase separating” protein is, is not trivial, as outlined, for example, in Farahi et al. (2021) Int J Mol Sci. 22: 3017. In this paper, different databases have been shown to present very different collections of phase-separating proteins, outlining the underlining cause that proteins may have very different roles in LLPS.

Reply: We thank the reviewer for making this salient point. We have included this citation in the manuscript and included a brief discussion on this topic.

3) FUS is studied in several different contexts, it is not clear why a particular mutant is used. Maybe it is related to disease, and its comparison to WT would have been very instructive in terms of disease mechanism?

Reply: We thank the reviewer for raising this question. We utilised this particular protein mutant since it is a well-studied system that is known to phase separate (Murakami et al., 2015; Patel et al., 2015). These citations have been added to the manuscript. The principal focus of the manuscript is to communicate the methodology, generalisability, scalability, and potential applications of the technique rather than draw conclusions on FUS mutation. Questions such as those raised regarding disease mechanism could be the subject of future study enabled by the PhaseScan method.

4) A deep underlying issue of these studies is that phase separation is not an equilibrium phenomenon, LLPS droplets are nucleated and evolve by Ostwald ripening. Therefore, observations by the PhaseScan system may depend on delay times typical for the system. This should be at least mentioned and discussed in the paper: maybe microfluidics enables to approach kinetic aspects of phase separation?

Reply: We agree with the reviewer that kinetic properties of phase separating systems are an important aspect to consider. We have revised the manuscript to include a more thorough discussion of the kinetic aspects of phase separation and included reference to time-dependent data (Figure S4), which shows that the phase diagram determined by PhaseScan remains unchanged with data acquisition between 5 – 60 min of droplet formation. This demonstrates that we are reporting the equilibrium phase boundary position. These results are in agreement with previous reports (Linsenmeier et al., 2019), which suggest that for droplet compartments with volume >100 pL, the characteristic timescale for protein condensate

formation is <1 min. Our measurement takes place >5 min after droplet encapsulation. Regarding kinetic experiments more generally, as shown in Figure 1h, dynamics of condensate merging events can be observed within the PhaseScan system, whilst Figure S7 demonstrates that condensate nucleation events can also be observed using our assay.

5) *It is central to the LLPS field that charge compensation by RNA often results in “reentrant” behaviour. The power of the system could be demonstrated if this was apparent (e.g., on Figure 2a, e or f).*

Reply: We agree that this is an important and characteristic phenomenon observed in protein phase separation. We have chosen to move the data for phase separation of PR25 (Fig 2) to the SI and have replaced it with a newly acquired phase diagram for heterotypic G3BP1/RNA phase separation which clearly demonstrates reentrant phase separation (Fig 2(d)). At a fixed concentration of G3BP1 (e.g., 1.75 μ M), the system transitions from homogeneous to phase separated and back to homogeneous as the RNA concentration is increased.

6) *When the effect of a small molecule (bis-ANS) is demonstrated, it would be very interesting to show that the technique enables screening of multiple conditions (compound concentration, or different compounds).*

Reply: We agree that these capabilities are a critical component for the potential future application of the technique in small-molecule screening. Within the manuscript, we have addressed the question regarding multiple compound concentrations in Figure 4, where we show the effect of a small molecule (1,6-hexanediol) on the phase separation propensity of the FUS/PEG system in a multi-dimensional manner, where the concentrations of small molecule, protein and molecular crowder are assessed in parallel.

7) *In conclusion, whereas phase diagrams of many data points are presented, one misses the point why this technique is superior to previous systems: a novel insight in at least one system, not apparent in previous lower-resolution approaches, should be demonstrated.*

Reply: We thank the reviewer for this comment. The problem we are addressing is that pre-existing methods do not allow practical generation of phase diagrams with high data density (thousands of data points) distributed along arbitrary dimensions in chemical space. We believe that our demonstrated ability to acquire phase diagrams at high throughput with independent dimensions will unlock further significant possibilities in the field, given the myriad of factors that should be investigated to characterise phase separating systems comprehensively.

References

- Bremer, A., Mittag, T., & Heymann, M. (2020). Microfluidic characterization of macromolecular liquid–liquid phase separation. *Lab on a Chip*, 20(22), 4225–4234. <https://doi.org/10.1039/D0LC00613K>
- Guillén-Boixet, J., Kopach, A., Holehouse, A. S., Wittmann, S., Jahnel, M., Schlüßler, R., Kim, K., Trussina, I. R. E. A., Wang, J., Mateju, D., Poser, I., Maharana, S., Ruer-Gruß, M., Richter, D., Zhang, X., Chang, Y. T., Guck, J., Honigmann, A., Mahamid, J., ... Franzmann, T. M. (2020). RNA-Induced Conformational Switching and Clustering of G3BP Drive Stress Granule Assembly by Condensation. *Cell*, 181(2), 346–361.e17. <https://doi.org/10.1016/j.cell.2020.03.049>
- Kopp, M. R. G., Linsenmeier, M., Hettich, B., Prantl, S., Stavarakis, S., Leroux, J.-C., & Arosio, P. (2020). Microfluidic Shrinking Droplet Concentrator for Analyte Detection and Phase Separation of Protein Solutions. *Analytical Chemistry*, 92(8), 5803–5812. <https://doi.org/10.1021/acs.analchem.9b05329>
- Li, P., Zeng, X., Li, S., Xiang, X., Chen, P., Li, Y., & Liu, B. F. (2022). Rapid Determination of Phase Diagrams for Biomolecular Liquid–Liquid Phase Separation with Microfluidics. *Analytical Chemistry*, 94(2), 687–694. https://doi.org/10.1021/ACS.ANALCHEM.1C02700/SUPPL_FILE/AC1C02700_SI_002.PDF
- Linsenmeier, M., Kopp, M. R. G., Grigolato, F., Emmanoulidis, L., Liu, D., Zürcher, D., Hondele, M., Weis, K., Capasso Palmiero, U., & Arosio, P. (2019). Dynamics of Synthetic Membraneless Organelles in Microfluidic Droplets. *Angewandte Chemie International Edition*, 58(41), 14489–14494. <https://doi.org/10.1002/anie.201907278>
- Murakami, T., Qamar, S., Lin, J. Q., Schierle, G. S. K., Rees, E., Miyashita, A., Costa, A. R., Dodd, R. B., Chan, F. T. S., Michel, C. H., Kronenberg-Versteeg, D., Li, Y., Yang, S. P., Wakutani, Y., Meadows, W., Ferry, R. R., Dong, L., Tartaglia, G. G., Favrin, G., ... St George-Hyslop, P. (2015). ALS/FTD Mutation-Induced Phase Transition of FUS Liquid Droplets and Reversible Hydrogels into Irreversible Hydrogels Impairs RNP Granule Function. *Neuron*, 88(4), 678–690. <https://doi.org/10.1016/j.neuron.2015.10.030>
- Patel, A., Lee, H. O., Jawerth, L., Maharana, S., Jahnel, M., Hein, M. Y., Stoyanov, S., Mahamid, J., Saha, S., Franzmann, T. M., Pozniakovski, A., Poser, I., Maghelli, N., Royer, L. A., Weigert, M., Myers, E. W., Grill, S., Drechsel, D., Hyman, A. A., & Alberti, S. (2015). A Liquid-to-Solid Phase Transition of the ALS Protein FUS Accelerated by Disease Mutation. *Cell*, 162(5), 1066–1077. <https://doi.org/10.1016/j.cell.2015.07.047>
- Selimović, E., Gobeaux, F., & Fraden, S. (2010). Mapping and manipulating temperature–concentration phase diagrams using microfluidics. *Lab on a Chip*, 10(13), 1696–1699. <https://doi.org/10.1039/B925661J>
- Yang, P., Mathieu, C., Kolaitis, R. M., Zhang, P., Messing, J., Yurtsever, U., Yang, Z., Wu, J., Li, Y., Pan, Q., Yu, J., Martin, E. W., Mittag, T., Kim, H. J., & Taylor, J. P. (2020). G3BP1 Is a Tunable Switch that Triggers Phase Separation to Assemble Stress Granules. *Cell*, 181(2), 325–345.e28. <https://doi.org/10.1016/j.cell.2020.03.046>

Reviewers' Comments:

Reviewer #1:

Remarks to the Author:

The present MS has been somewhat improved upon revision. However, unfortunately, the revisions have not addressed any of my 2 major concerns (points 1 and 2 in my original review) in sufficient detail. With all due respect, I am unable to see the novelty or originality of "phasescan" to a level that would warrant publication in this highly prestigious journal on the basis of this technology alone, as the authors argue. I understand the points made with regard to feasibility, generalizability, and scalability, but my concern expressed in point # 1 in the original review remains. A potential solution to my concerns is the following: Can the authors show how "phasescan" platform leads to significant new insights into the phase diagrams of biomolecular condensates, as I originally requested? Therefore, I would earnestly request (again) the authors to perform experiments that would lead to some new insights into at least one phase separating system in addition to all the proof-of-principle data shown in this MS. Given all the advantages of this microfluidic platform, I don't think it is an unreasonable request in terms of the experimental feasibility. The addition of G3BP1-RNA data is good, but again, we are not really learning anything new that we did not know already. I note that This particular point was also highlighted by Reviewer 3 as well (point # 7).

Reviewer #2:

Remarks to the Author:

The authors have done a very good job of addressing the earlier comments. The work and method developments will be of substantial interest and utility to the scientific community in this exciting and rapidly expanding research area. Therefore, I believe the manuscript can now be published.

Reviewer #3:

Remarks to the Author:

I have read the detailed answers given, and action taken, in response to my points raised on the first version of the manuscript. I think the questions have been appropriately handled and the manuscript is in a good shape now for acceptance.

Reviewer #1:

The present MS has been somewhat improved upon revision. However, unfortunately, the revisions have not addressed any of my 2 major concerns (points 1 and 2 in my original review) in sufficient detail. With all due respect, I am unable to see the novelty or originality of "phasescan" to a level that would warrant publication in this highly prestigious journal on the basis of this technology alone, as the authors argue. I understand the points made with regard to feasibility, generalizability, and scalability, but my concern expressed in point # 1 in the original review remains. A potential solution to my concerns is the following: Can the authors show how "phasescan" platform leads to significant new insights into the phase diagrams of biomolecular condensates, as I originally requested? Therefore, I would earnestly request (again) the authors to perform experiments that would lead to some new insights into at least one phase separating system in addition to all the proof-of-principle data shown in this MS. Given all the advantages of this microfluidic platform, I don't think it is an unreasonable request in terms of the experimental feasibility. The addition of G3BP1-RNA data is good, but again, we are not really learning anything new that we did not know already. I note that this particular point was also highlighted by Reviewer 3 as well (point # 7).

Reply: We thank the reviewer for the overall positive feedback on our revised manuscript. With regard to the remaining question, the focus of our manuscript is the demonstration of PhaseScan as an enabling technology. Although phase diagrams are a primary method for biophysicists seeking to characterise phase separating systems, phase diagrams generated experimentally are semi-quantitative at best. PhaseScan enables rigorous, quantitative characterisation and comparison of condensate systems, and represents a significant advance in solving one of the primary experimental challenges in this rapidly growing field.

The comprehensive range of condensate systems we validate in the manuscript demonstrates the versatility and general applicability of the technique. We expect future studies to use this technology to generate novel insights into specific systems. Indeed, this is already happening, and phase diagrams of biomolecular condensates have been shown in recent work to provide crucial mechanistic information regarding chromatin remodelling (<https://doi.org/10.1101/2021.10.14.464384>) and antibody phase separation (<https://doi.org/10.1101/2022.03.07.483238>).

Reviewer #2:

The authors have done a very good job of addressing the earlier comments. The work and method developments will be of substantial interest and utility to the scientific community in this exciting and rapidly expanding research area. Therefore, I believe the manuscript can now be published.

Reply: We thank the reviewer for their positive feedback on our revised manuscript and for the recommendation to publish the paper.

Reviewer #3:

I have read the detailed answers given, and action taken, in response to my points raised on the first version of the manuscript. I think the questions have been appropriately handled and the manuscript is in a good shape now for acceptance.

Reply: We appreciate the enthusiastic feedback and the recommendation to publish the paper.